# The Evolving Concept of the Multidisciplinary Approach in the Diagnosis and Management of Interstitial Lung Diseases

**DOI:** 10.3390/diagnostics13142437

**Published:** 2023-07-21

**Authors:** Stefano Sanduzzi Zamparelli, Alessandro Sanduzzi Zamparelli, Marialuisa Bocchino

**Affiliations:** 1Division of Pneumology, A. Cardarelli Hospital, 80131 Naples, Italy; stefanosanduzzi@gmail.com; 2Department of Clinical Medicine and Surgery, Section of Respiratory Diseases, University Federico II, Azienda Ospedaliera dei Colli-Monaldi Hospital, 80131 Naples, Italy; marialuisa.bocchino@unina.it; 3Staff of UNESCO Chair for Health Education and Sustainable Development, University Federico II, 80131 Naples, Italy

**Keywords:** interstitial lung diseases, idiopathic pulmonary fibrosis, hypersensitive pneumonia, connective tissue disease–interstitial lung diseases, idiopathic pneumonia with autoimmune features, multidisciplinary diagnosis, multidisciplinary team

## Abstract

Background: Interstitial lung diseases (ILDs) are a group of heterogeneous diseases characterized by inflammation and/or fibrosis of the lung interstitium, leading to a wide range of clinical manifestations and outcomes. Over the years, the literature has demonstrated the increased diagnostic accuracy and confidence associated with a multidisciplinary approach (MDA) in assessing diseases involving lung parenchyma. This approach was recently emphasized by the latest guidelines from the American Thoracic Society, European Respiratory Society, Japanese Respiratory Society, and Latin American Thoracic Association for the diagnosis of ILDs. Methods: In this review, we will discuss the role, composition, and timing of multidisciplinary diagnosis (MDD) concerning idiopathic pulmonary fibrosis, connective tissue disease associated with ILDs, hypersensitive pneumonia, and idiopathic pneumonia with autoimmune features, based on the latest recommendations for their diagnosis. Results: The integration of clinical, radiological, histopathological, and, often, serological data is crucial in the early identification and management of ILDs, improving patient outcomes. Based on the recent endorsement of transbronchial cryo-biopsy in idiopathic pulmonary fibrosis guidelines, an MDA helps guide the choice of the sampling technique, obtaining the maximum diagnostic performance, and avoiding the execution of more invasive procedures such as a surgical lung biopsy. A multidisciplinary team should include pulmonologists, radiologists, pathologists, and, often, rheumatologists, being assembled regularly to achieve a consensus diagnosis and to review cases in light of new features. Conclusions: The literature highlighted that an MDA is essential to improve the accuracy and reliability of ILD diagnosis, allowing for the early optimization of therapy and reducing the need for invasive procedures. The multidisciplinary diagnosis of ILDs is an ongoing and dynamic process, often referred to as a “working diagnosis”, involving the progressive integration and re-evaluation of clinical, radiological, and histological features.

## 1. Introduction

Interstitial lung diseases (ILDs) represent a huge and heterogenous group of pulmonary disorders, characterized by varying degrees of inflammation and fibrosis of the lung with different prognoses and management options. Despite extensive analysis, the diagnosis of ILDs remains challenging, as the specific etiology cannot be identified in 10–20% of cases, due to the common respiratory clinic and the frequent overlap of radiologic and/or histopathologic patterns among various diseases [1]. Although idiopathic pulmonary fibrosis (IPF) is the prototype of a progressive fibrotic disease associated with a poor prognosis and premature mortality, it is associated with over 40% of non-IPF ILDs [2]. Based on this, the paramount significance of achieving an accurate and early diagnosis, along with the timely initiation of disease-specific therapy, becomes evident to alter the outcome [3,4].

Since 2001, the American Thoracic Society (ATS) and the European Respiratory Society (ERS) have introduced a critical change in the diagnostic process of ILDs, highlighting the importance of a multidisciplinary diagnosis (MDD) based on the integration of clinical, radiologic, and pathologic data [5]. A face-to-face discussion between different health professionals, traditionally represented by an ILD-expert respiratory physician, and other figures, such as radiologists and histopathologists, aims to use available clinical data to generate a consensus diagnosis with the highest possible level of accuracy.

Before this review, despite the literature demonstrating an exceedingly low inter-observer agreement between expert thoracic pathologists, histopathological evaluation was considered the gold standard for ILD diagnosis [6,7]. In addition to the low level of consensus on the histological sample between different doctors, the main limitation of this kind of diagnosis was linked to the extreme variability of the histological data, concerning the representativity of the sample taken strongly depending on the operator’s experience. In this context, out of 133 lung biopsies taken from 83 ILD patients, the Nicholson et al. study [8] showed a poor inter-observer agreement among 10 thoracic pathologists, with a 0.38 kappa coefficient of agreement (κ), making a 100% confidence diagnosis in only 39% of cases. Notably, more than 50% of the inter-observer variation was related to the differential diagnosis between nonspecific interstitial pneumonia (NSIP) and usual interstitial pneumonia (UIP). The uncertainty linked to the histopathologic data is at a maximum in NSIP patterns because of the presence of different pathological patterns in different lobes in the same patient. Moreover, divergent pathological diagnoses can be often caused by surgical lung biopsy (SLB) sampling error [9]. In this context, Flaherty et al. [10] demonstrated that a histopathological diagnosis based on a single lung biopsy site can be inaccurate, in particular about NSIP/UIP differentiation, with around 26% of patients with discordant diagnoses. Furthermore, histopathological diagnosis is not fixed, but it can be modified in 20% of cases, particularly when an intermediate pattern is present [11]. It is important to bear in mind that even a histopathologic UIP pattern is not surely related to IPF, as demonstrated in Tominaga’s study [12]. In this study, the integration of clinical and radiological data from 95 IPF patients confirmed by a histological pattern compatible with UIP led to a progressive reduction in post-MDD IPF diagnosis.

More recently, MDA was first emphasized by the 2013 ATS/ERS update for interstitial idiopathic pneumonia’s (IIP’s) [13] management and further reinforced by the 2018 Fleischner statement [14]. These guidelines proposed updated IPF diagnostic criteria, endorsing the formulation of an MDD of IPF, mainly in the absence of definitive radiological or histopathological findings. Based on the literature, which has shown a variable frequency of IPF ranging from 60 to 90% for patients with probable UIP patterns, MDD is mandatory in all ILD patients who lack a definite UIP pattern on high-resolution chest tomography (HRCT). The impact of this approach is supported by Kondoh’s study [15], where out of 179 patients with probable HRCT UIP patterns, an IPF diagnosis was established in 50% of cases following MDD.

The growing confidence in MDD as a highly accurate assessment of ILDs was emphasized in the 2021 guidelines from the ATS, ERS, Japanese Respiratory Society (JRS), and Latin American Thoracic Association (ALAT) [16]. In addition to its role in determining the need for biopsy investigations and avoiding invasive procedures—UIP or probable UIP patterns on imaging in the appropriate clinical setting is enough to make a diagnosis of IPF—MDD can also aid in selecting the most appropriate sampling tool for lung parenchyma, such as transbronchial cryo-biopsy (TBLC) or SLB. According to this guideline [16], a biopsy is generally considered when an indeterminate or inconsistent HRCT UIP pattern conflicts with clinical data. Data from the literature suggest restricting the use of MDD to more complex diagnostic cases, with a focus on clinician consensus when the diagnostic IPF criteria are more evident—UIP on HRCT, a rapidly progressive disease course, and no identifiable triggers. This approach, as emphasized by the retrospective evaluation of 318 ILD patients made by Chaudhuri et al. [17], is associated with the potential to change the diagnosis.

In contemporary practice, multidisciplinary assessment (MDA) has become the standard approach for evaluating ILDs due to its high level of diagnostic confidence and inter-observer agreement. Although initially developed for IIP’s diagnosis, an MDA is now widely used as the standard in the evaluation of ILDs, not only those that are idiopathic but also those related to an underlying connective tissue disease (CTD) or linked with autoimmune features such as idiopathic pneumonia with autoimmune features (IPAF) [11]. In contrast to an IPF diagnosis, many ILDs are not covered by evidence-based diagnostic guidelines; therefore, a level of disagreement in MDD is predictable [13]. Furthermore, apart from assessing new cases, MDT discussions are key for reconsidering a previous diagnosis according to disease behavior and response to therapy.

## 2. MDD Has a Crucial Role in Guiding the Choice of Lung Tissue Sampling Procedures

In the most challenging cases, when histological data are required, the latest ERS guideline [16] emphasizes the role of MDD in guiding the choice of biopsy technique and in the evaluation of samples. Based on European epidemiological studies [18,19,20], it has been found that 28–38% of IPF patients were diagnosed using SLB. However, in cases where SLB is not feasible, TBLC is a valid alternative. The latter is a method capable of obtaining lung tissue samples due to the use of a cryoprobe—either 1.9 mm or 2.4 mm—and, thanks to the Joule–Thomson principle, this procedure is performed [21] in intubated patients with deep sedation or general anesthesia, has a high diagnostic yield (>70%), and an excellent safety profile (adverse event rate lower than 20% and negligible mortality). TBLC is a technique well-tolerated by patients, and is capable of providing large biopsy specimens, leading to high-confidence diagnoses of UIP in about 50% of cases [22,23]. The literature highlights the impact of integrating TBLC with MDT data, which increases the high-confidence diagnosis of IPF from 16% to 63%, changing the first clinical-radiological diagnosis in 26% of cases. Agreement between TBLC and SLB has been reported ranging from 38% to 70.8%, but it can improve with the number of samples taken [24]. As postulated in the latest ERS guidelines on TBLC in the diagnosis of ILDs [21], TBLC is recommended as the preferred method when MDT evaluation requires the integration of histopathological data. If the first sample is uninformative, rather than a repetition of it, it is indicated to perform an SLB as an add-on test. In addition, it is recommended to discuss MDD before any type of procedure to plan the biopsy and to evaluate the sample with a higher grade of accuracy. Although TBLC has a lower diagnostic yield compared with SLB, the latest TBLC guideline suggests this method as the first step in ILD diagnosis due to its lower adverse event rates, shorter length of hospitalization, and lower costs, particularly if it is performed at experienced centers. It is important to note that TBLC should be performed at expert centers due to the potential risk of complications such as pneumothorax, bleeding, or exacerbation of IPF in some classes of patients—such as massive emphysema, pulmonary hypertension, and upper localization of lesions [1,2]. The higher level of diagnostic accuracy is associated with the sampling procedure, particularly when executed from at least two different sites—different segments of the same lobe or different lobes. This approach increases the chance of identifying the ILD’s histologic heterogeneity, with a success rate of 92–96% when two samples are taken. However, it is important to note that performing biopsies from multiple sites also doubles the risk of pneumothorax—that at two sites is 24.6% vs. that at one site is 15.2%—and tanking samples close to the pleura area (within 1 cm), where the pathologic changes of IPF are most prominent [24]. TBLC increases its utility when implemented by MDD, enabling the discussion of difficult cases and the interpretation of histological samples by different experts including pulmonologists, radiologists, and rheumatologists to improve the possibility of ILD diagnosis. Although TBLC is not as effective as SLB in obtaining peripheral specimens, its diagnostic yield is comparable to that of SLB, because subpleural and/or para-septal fibrosis are not critical criteria in making an IPF diagnosis [25]. However, despite its growing diffusion and expertise, skepticism surrounding TBLC persists due to the lack of standardization and the need for larger histologic samples for studying pathogenesis and molecular patterns [14,26].

When less invasive approaches fail, SLB is considered the reference standard. Typically obtained thoracoscopically, SLB leads to larger samples that contain peripheral structures of the secondary pulmonary lobule, resulting in a diagnostic yield of 90% [6,7]. In addition to its significant costs and risks, including a mortality rate of around 2%, the use of SLB is limited in advanced stages of the disease, elderly patients, and patients with multiple comorbidities [1,2].

According to Montufar et al. [27] and Troy et al. [28], TBLC is suggested as a method of choice compared with SLB, which remains the gold standard, especially when applied in the context of MDD. On the other hand, the lack of a standard technique for TBLC has led Lynch et al. [14] to consider SLB as a histologic tool for ILD diagnosis. Given the heterogenous spectrum of ILDs, it is necessary to assess the optimal diagnosis approach for different sampling modalities that need to be tailored to specific patients. An insightful approach is proposed by Castillo et al. [29], suggesting the selection between the two sampling techniques based on the clinical context. When HRCT central findings are observed, patients’ lung function is compromised, and the bronchoscopy team has high expertise, TBLC is recommended as a first option. Conversely, if peripherical lesions are present, lung function is preserved, and a higher diagnostic yield is required, SLB should be the first to be considered. Ultimately, it is also critical to consider the patient’s point of view; their preference may lean towards undergoing an endoscopic procedure rather than a surgical one. In the attempt of making an ILD diagnosis, instead of finding the best method to obtain lung samples, it is critical to determine the best contest which uses different techniques. The existing literature on SLB vs. TBLC still presents conflicting evidence, as both approaches have their advantages and limitations.

When TBLC or SLB cannot be performed, histological diagnosis can be obtained via video-assisted thoracic surgery (VATS) biopsy, which is an extremely invasive technique requiring endotracheal intubation and mechanical ventilation. This sampling approach, chosen based on multidisciplinary agreement, is associated with high morbidity (up to 30%) and mortality rate (up to 4%), particularly in elderly patients [30]. In recent years, non-intubated VATS biopsy performed in awake subjects under loco-regional anesthesia has slowly emerged as a safety technique with minimal complications for patients with undetermined ILDs. This approach, compared with classical VATS, has a lower ratio of complications (odds ratio for postoperative complications: 8.1, *p* = 0.011) and faster recovery times. Also, it is associated with the use of anesthetics different than those for muscle relaxants, the non-use of double-lumen tube intubation, and positive pressure ventilation injuries [31]. In a recent review by Rossi et al. [32], which aimed to analyze the differences between the two approaches, no statistically significant differences were found regarding the identification of histopathological features. Although the width of the biopsies is significantly deeper with traditional VATS (31.5 mm vs. 25.6 mm; *p* = 0.01), both techniques have shown a diagnostic yield close to 100% (same biopsy length, average number of sampled lobes, and mean number of slides). By contrast, awake VATS was characterized by a higher level of diagnostic confidence (100% vs. 75%; *p* = 0.007) and an older cohort of patients (69.5 vs. 64.5 years old; *p* = 0.02). In these cases, the choice among the two approaches depends on the level of expertise of the thoracic surgeons and anesthesiologists within the MDT. This evaluation, without sacrificing diagnostic performance, should be considered a careful evaluation of the risk–benefit ratio for every single patient, taking into account the disease severity, patient comorbidities, and the increased mortality associated with potential acute exacerbations related to intubation. However, due to the limited diffusion of the awake VATS, further studies are necessary to validate its safety and efficacy.

On the basis of the data published by the INPULSIS study [4], which analyzed the effect of nintedanib in reducing lung function decline in IPF patients, more recent studies such as the INBUILD and SENSCIS trials [33,34] investigated the benefits of the use of antifibrotic agents across a range of progressive fibrotic lung diseases (PF-ILDs), including unclassifiable ILD, fibrotic hypersensitive pneumonia (HP), fibrotic NSIP, IPAF, CTD-ILD, and other less common fibrotic variants. PPF-ILDs represent a subset of fibrotic ILDs sharing with IPF the disease’s natural course, characterized by radiological, functional, and clinical evidence of progression over time despite therapy. According to the INBUILD trial [33] and the 2022 ATS/ERS/ALAT/JRS IPF update [16], the definition of a PF-ILD is an ILD other than IPF that has radiological evidence of pulmonary fibrosis, with at least two of the following three criteria occurring within the past year: (i) worsening respiratory symptoms, (ii) decline in forced vital capacity (FVC) ≥5 or diffusing capacity of the lungs for carbon monoxide (DLCO) ≥10%, (iii) and/or radiological evidence of disease progression.

Despite the PF-ILD usually requiring antifibrotic therapy, biopsy should also be performed in an attempt to differentiate ILD subtypes, identifying patterns more prone to progression, leading to earlier initiation of therapy, and helping to choose between starting antifibrotic or immunosuppressive therapy in cases where both might be indicated, such as fibrotic HP. According to the prognostic value of lung biopsy in the identification of PF-ILDs, TBLC represents a valid alternative to SLB if performed at skilled centers, due to its significantly lower morbidity and mortality [35].

Based on these findings, Wells et al. [36] highlighted the value of ‘lumping’ IPF with other forms of PF-ILDs, in opposition to a ‘splitting’ approach with the ultimate goal of precision medicine: in this context, it is still challenging to choose when to perform a biopsy, considering each patient individually, mainly when MDD is not available.

## 3. Genomic Classifier and Its Role in Supporting MDD

Another topic covered by the latest IPF guidelines [16] was the recent introduction of genomic classifier testing (GCT) to clinical use. This technique utilizes machine learning to analyze the whole-transcriptome RNA sequencing and the gene expression patterns in lung tissue samples obtained by transbronchial biopsy, thereby improving the diagnostic yield of ILD [37,38]. By categorizing the presence or absence of diagnostic features, GCT enables distinguishing UIP from non-UIP histopathology. This supports the MDT in making a more confident diagnosis of IPF in patients without a definite UIP pattern on chest imaging.

A recent systematic literature review [39] estimated that GCT can differentiate UIP and non-UIP histopathology with a sensitivity of 68% (95% Confidence Interval (CI), 55–73%) and a specificity of 92% (95% CI, 81–95%). This provides valuable support to physicians and MDTs in confirming the diagnosis of UIP in patients without a definitive radiologic pattern for UIP. Opposite to the diagnostic confidence improvement of a UIP histopathology-predicted pattern using GCT, non-UIP histopathology prediction may require further confirmation through SLB or TBLC due to the frequent occurrence of false-negative results. Higher confidence in the diagnostic evaluation of ILD can be implemented via the integration of GCT results with clinical and radiologic information in an MDD context. Two different studies [38,40] have shown that the diagnostic accuracy of GCT increases from 56% to 89% (agreement, k = 0.64) and from 43% to 93% (agreement, k = 0.75) after MDT evaluation.

The integration of GCT into the MDD process increases diagnosis confidence levels, particularly in cases where chest imaging shows an indeterminate UIP pattern or probable UIP with confounding clinical factors—such as autoantibodies, ILD-associated medication, and a history of environmental or work exposure. This integration increases the proportion of IPF cases diagnosed from 31% to 92%. In a study by Kheir et al. [40], the addition of GCT to the clinical and radiological data resulted in a mild increase ranging from 17% to 29% in the proportion of high-confidence diagnoses, despite a high overall agreement of 92% between the GCT and the final MDD of UIP or non-UIP. The combination of GCT with TBLC can further improve the diagnostic yield in patients with a probable UIP pattern on HRCT imaging, while TBLC alone may be used in cases with a confident non-UIP diagnosis. However, GCT implementation appears to be less beneficial when the HRCT scan shows patterns inconsistent with UIP.

While additional studies are needed to define its precise accuracy due to the high frequency of false-negative results, GCT provides important diagnostic information. When integrated with clinical and radiological elements, it may reduce the need for additional and more invasive sampling as SLB or TBLC. A review conducted by Richeldi et al. [41], which analyzed the impact of radiological data integration in reducing false-negative results, reported that GCT sensitivity increased from 60.3% to 79.2% when in conjunction with the HRCT pattern of UIP.

## 4. MDD Inter-Observer Agreement

Since its institution in the assessment of ILDs, the MDA has been associated with increased levels of diagnostic confidence and improved inter-observer agreement. The effectiveness of this approach, which involves progressive improvement of a provisional diagnosis through the integration of different data, has already been investigated in the literature.

Flaherty et al. [11] conducted a study to assess the impact of an integrated expert approach in ILD and were the first to demonstrate that the integration of clinical, radiological, and pathological data increases confidence in IPF diagnosis and improves inter-observer agreement (0.39 vs. 0.88) compared with the first approach. Among all ILDs, IPF has the highest level of diagnostic agreement between community and academic physicians.

Although limited to IPF diagnosis, Thomeer et al. [42] demonstrated that accuracy in diagnosis is improved by higher levels of agreement between clinicians, even from different European centers, improving accuracy at 87.2%.

In recent decades, due to the crescent development in radiology, HRCT has gradually contributed an improved impact on MDD. In a study by Aziz et al. [43] involving 168 ILD patients, after integrating HRCT data, they found that the clinician’s first-choice diagnosis changed in 51% of cases, improving diagnostic accuracy. The diagnostic role of HRCT was also confirmed by Travis’s [44] study, in which, among 105 biopsy-proven NSIP patients, 21 were disconfirmed after merging radiological data.

As mentioned earlier, although the literature shows that MDD improves diagnostic agreement and accuracy in clinical practice compared with individual diagnoses made by clinicians, radiologists, and histopathologists, there is still no established reference standard. In his study, Walsh [45] highlighted the existence of significant discordance between different MDTs that evaluated the same clinical, imaging, and histopathological data [11,42]. Despite the dynamic integration of clinical, radiological, and pathological data in the MDD process leading to improved diagnostic accuracy, Walsh’s study [45] did not observe this improvement. This discrepancy may be attributed to the fact that diagnostic accuracy was more frequently obtained by radiologists and pathologists who, in that particular study, did not have access to clinical data. Consequently, the diagnostic process relied on pattern interpretation, resulting in increased confidence but not necessarily improved accuracy [11].

## 5. MDD as a Tool in the Dynamic Process of Diagnosis

The 2018 Fleischner Society statement [14] endorsed the concept of progressive and dynamic integration of clinical, radiological, and pathological data, emphasizing the need to re-evaluate previous diagnoses based on disease progression and therapy response, which may lead to changes to the initial diagnosis. The available literature [11,46] has shown that MDD results in a change in diagnosis in 40% to 53% of cases. Ageely et al. [47] confirmed that MDD provided a new diagnosis or modified a preexisting diagnosis in 37% of the cases when compared with the diagnosis pre-MDD. The study reported a 41% concordance rate between pre-MDD and MDD, with the highest level of agreement observed in IPF diagnosis (81%). In contrast to IPF, due to the lack of consensus of diagnostic criteria and evidence-based guidelines, the concordance rate for HP diagnosis was low, and a relatively high proportion of cases referred to as HP were previously deemed as unclassifiable ILD.

In this context, the concept of a “working diagnosis” has emerged, allowing for the justification of disease-specific therapy even in the absence of a definite diagnosis, re-evaluating the case in the following controls [48]. While strict adherence to guidelines maximizes diagnostic accuracy, in clinical practice it may lead to delays in diagnosis and treatment when dealing with unclassifiable cases. Nowadays, if diagnostic criteria are not fully met for a confident ILD diagnosis, the integration of data and expert evaluation may result in a provisional diagnosis with a high level of confidence, guiding the diagnostic and therapeutic process. Cases that cannot reach a confident or provisional diagnosis even after expert MDD are considered unclassifiable ILD, a category endorsed in the 2013 updated classification [13]. The incidence rates of unclassifiable ILD vary from 10% to 44%, reflecting large variability that can be explained by inconsistent definitions [46,49]. In Ageely et al.’s study [47], it was found that 12% of the patients who did not fulfill diagnostic criteria received a provisional diagnosis that was useful to their management. In 21% of cases, ILD was defined as unclassifiable due to a lack of histopathological data at first assessment, being diagnosed after MDD review by a subsequent biopsy.

To standardize the heterogeneous terminology, Ryerson et al. [50] proposed the subclassification of the ILD diagnosis probability into “confident”, “provisional”, and “unclassifiable ILD” categories based on the degree of diagnostic confidence (>90%, between 89 and 50%, and <50%, respectively). The implications of using these terms on the decision to perform a biopsy or initiate a specific therapy were shown in an international study involving 404 clinicians who evaluated 60 cases of suspected IPF. The findings revealed an attitude that deviates from current guidelines. In cases where there was a provisional high level of confidence in the IPF diagnosis, only a minority of patients (29.6%) underwent SLB. Instead of a definite diagnosis as defined by current guidelines, a “working diagnosis” approach was adopted, leading to the prescription of antifibrotic therapy without performing a biopsy in 63% of patients with a diagnostic likelihood of 70%, as well in 63% and 41.5% of cases with provisional high confidence and low confidence IPF diagnoses, respectively [51]. Around 10% of patients presenting with a not-definite HRCT UIP pattern cannot undergo SLB due to factors such as advanced age, advanced disease, or poor clinical conditions. These patients do not fall into any specific diagnostic category and, therefore, do not have access to the available treatments. In such cases, less-invasive procedures like TBLC are available, providing high diagnostic confidence when performed by skilled physicians [52].

In some cases, more than one MDD meeting is needed to reach a diagnosis, as reported by De Sadeleer et al. [46]. The literature data suggest that considering atypical cases as “not yet classified”, rather than unclassifiable ILD, as a definite diagnosis may only be achieved through clinical, radiologic, and histopathologic data. This highlights the importance of a dynamic approach to ILD patients. An exception is represented by cases in which a specific guideline-based diagnosis may never be achieved, such as if SLB is contraindicated due to factors such as advanced age, advanced disease, or comorbidities. This emphasizes the importance of early referral to centers with high ILD expertise, particularly for patients with initial unclassifiable diagnoses who require additional diagnostic tests and lung biopsy. MDD meetings also play a crucial role in changing patient management, including the initiation or discontinuation of therapy, as shown in 39% of patients in the Ageely study [47]. It is worth noting that management was changed in 46% of patients, even for those with high-concordant pre-MDD and MDD. This emphasizes the central role of expert opinion not only in making a diagnosis, but also in drawing up an appropriate treatment—such as anti-fibrotic drugs for IPF or progressive fibrotic ILDs and immunosuppressive/anti-inflammatory therapy for non-fibrotic ones.

The introduction of disease behavior in the diagnostic process within guidelines obliges the MDD group to revise, confirm, and, in boundary cases, reformulate diagnoses, leading to increased confidence [14]. Although ILD diagnosis can change over time owing to emerging clinical or serological results, there are only a few studies that have analyzed the role of MDD in follow-up and response to therapy. In a retrospective study [53] that evaluated 56 patients over 7 months of follow-up, it was found that dynamic re-evaluation of new clinical data or HRCT patterns can modify the initial MDD diagnosis in 10.7% of cases, or the level of agreement in 25% of cases, highlighting the need for continued expert meetings during the follow-up phase. Despite the growing importance of MDD, the Australian IPF Registry (AIPFR) [54] has shown that in 23% of cases, IPF guidelines were not followed by referring physicians. Following MDD of 93 ILD patients, the diagnosis was changed in 53% of cases, and 71% of unclassifiable diseases were reclassified, significantly impacting the therapeutic approach and leading to an increase in antifibrotic drugs.

While MDD is globally recognized for its utility in making a final diagnosis, there is no reference standard available to measure its validity. Notably, the majority of MDD evaluations concern only the diagnosis aspect, whereas therapeutic and disease management decisions are still often made by a single physician, typically the clinician who resolved the diagnostic uncertainty, and, frequently, is the pulmonologist alone [55,56].

## 6. MDD’s Role in Evaluating Prognosis

The role of MDD in evaluating prognosis was demonstrated by De Sadeleer [46] in a large study involving 938 patients, where the diagnosis was reached in 79.5% of cases, modified in 41.9% of cases after MDD, and not achieved only in 19.5% of the patients. This suggests that further investigations were required in 16% of the total cohort. In this study, IPF patients demonstrated a worse prognosis than those diagnosed as non-IPF after MDD [Hazard ratio (HR) 4.31, *p* < 0.001]. However, patients who were initially identified as IPF but had a change in their diagnosis after expert discussion showed a better prognosis compared with those diagnosed as IPF (HR 0.37, *p* = 0.094) [43]. In another study by Nakamura et al. [57], among 33 patients initially identified as unclassifiable-ILD, MDD confirmed the initial diagnosis in 54.5% of cases, changing the diagnosis in the remaining ones. The IFIGENIA trial [42], a randomized placebo-controlled trial conducted on IPF patients, in which N-Acetylcysteine was associated with azathioprine plus steroid therapy, cases diagnosed as IPF by clinicians alone were disproved in 12.8% of cases after discussion with thoracic radiologists and expert pathologists.

The literature indicates that MDD is considered effective at achieving a first-choice diagnosis for both IPF and CTD-ILD with good agreement (κ = 0.60 and κ = 0.64, respectively), but fair for idiopathic NSIP and HP (κ = 0.25 and κ = 0.24, respectively). However, this lack of confidence is related to the absence of defined classification criteria available for such diseases [45]. As mentioned before, besides the diagnostic process, MDD should be performed to evaluate disease prognosis in particular conditions. For example, in a study involving 47 IPF patients who underwent MDD including SLB, the integration of different data allowed for the identification of negative prognostic factors for survival [58]. MDD can also be applied to select the appropriate drug and related timing therapy. In a cohort of patients with CTD-ILD, Kalluri [59] highlighted the higher level of clinician participation in managing their disease compared with IPF patients who received care within a single setting.

Table 1 shows the evidence summarized of how MDD can change the initial diagnosis by integrating clinical, radiological, and pathological data.

## 7. MDD in Evaluating HP

Among all ILDs, HP is particularly challenging to diagnose, as it requires the integration of anamnestic, clinical, radiological, and histopathological data. HP is an immune-mediated disease that affects the lungs of susceptible individuals after repeated exposure in time to identified or unidentified inciting agents, such as organic powders and low-molecular-weight chemical particles. Differentiating HP from other ILDs can be extremely challenging due to the highly variable clinical, radiological, and histopathological features of HP, as well as the overlap with those belonging to other ILDs, and shearing features of other acute and chronic pulmonary diseases [62]. The identification of clear inhalation exposure, in particular in cases of fibrotic/chronic disease, can be challenging. Therefore, the latest ATS/JRS/ALAT HP guidelines [63] emphasize the role of an MDT that includes a clinician, a radiologist, and a pathologist, who work together to integrate data from multiple domains to achieve an accurate diagnosis.

The diagnosis of HP requires a multidisciplinary approach involving clinical, radiological, and, in some cases, lymphocytosis on bronchoalveolar lavage (BAL) fluid or histopathological data. In addition to the CHEST guidelines [64] criteria for HP diagnosis—which include a clinical history of exposure to a potential antigen and a typical HP pattern on HRCT scan—according to the ATS/JRS/ALAT guidelines [63] a confident diagnosis of HP requires evidence of BAL lymphocytosis. Both guidelines emphasize the critical role of MDD in the diagnostic management of HP via the integration of clinical, anamnestic, radiologic, and BAL data. The ATS/JRS/ALAT guidelines [63] highlight that BAL should be performed along with exposure history and an HRCT scan before MDD. On the other hand, the CHEST guidelines [64] suggest performing this technique only after an MDD of clinical and radiological data, avoiding performing it in those cases that have a high pre-test probability of HP. In contrast to typical HP cases, in which a confident diagnosis can be achieved by avoiding the lung biopsy performance, in patients who have a suspicion of HP but atypical features a lung sample is suggested, considering the individual risk–benefit ratio. Particularly due to the extensive overlap between fibrotic HP and IPF, it is challenging to achieve a confident diagnosis and consequent early management. Guler et al. [65] have highlighted the validity of MDD as the reference standard for the diagnosis of fibrotic HP. MDD is associated with a stronger prognostic value and plays a crucial role in diagnostic decision-making in complex cases.

## 8. MDD in Evaluating CTD-ILDs

CTDs, including rheumatoid arthritis (RA), systemic lupus erythematosus (SLE), inflammatory idiopathic myopathies (IIMs), Sjögren’s syndrome (SS), systemic sclerosis (SSc), and mixed connective tissue disease (MCTD), encompass a wide spectrum of systemic autoimmune disorders that share common underlying mechanisms of pathogenesis. Due to their frequent multiorgan involvement, each CTD may show lung parenchymal damage before or during the disease. This is particularly notable in myositis spectrum disorders such as any-synthetase syndrome (ASS), where ILD may manifest as the predominant or sole feature in 10–30% of cases of ILD [66]. However, the identification of the etiology at the base of this lung involvement is extremely difficult due to the overlapping clinical characteristics, serology, and radio-histological patterns. Despite some radiological patterns such as NSIP or organizing pneumonia (OP) being more typical of CTD-ILD, a definite UIP pattern does not exclude an underlying autoimmune disease, which is frequently observed in RA and rarely in SSc [67].

Although there are no definite recommendations for CTD-ILD diagnosis due to its highly heterogeneous presentation, the first attempt to standardize a diagnostic process was made by the Thoracic Society of Australia and New Zealand [68]. They introduced the concept of an MDA involving a respiratory clinician, radiologist, pathologist, and rheumatologist to achieve the best possible outcome by avoiding invasive procedures such as lung sampling and increasing diagnostic confidence. As highlighted by the study of Møller et al. [69], an MDA is crucial to establishing a confident diagnosis in CTD-ILDs, in particular in those cases presenting with an NSIP pattern. Their retrospective analysis demonstrated that the diagnosis was changed in a statistically significant number of patients after MDD evaluation, leading to the classification of 25.8% of cases previously identified as iNSIP.

The ATS/ERS/JRS/ALAT guidelines [16] for IPF diagnosis have reinforced the concept of an MDD evaluation for ILDs. These guidelines have emphasized the need to exclude CTD-related lung involvement and recommend the involvement of radiologists, pathologists, pulmonologists, and, if necessary, rheumatologists, in the MDT to improve the diagnostic agreement. Notably, the current IPF guidelines routinely recommend performing the independent serological test as C-reactive protein (CRP), erythrocyte sedimentation rate (ESR), antinuclear antibodies (ANA), rheumatoid factor (RF), myositis panel, and anti-cyclic citrullinated peptide (ACPA), and consulting a rheumatologist in the case of positivity of those serological tests or if there are clinical manifestations suggesting an underlying rheumatological disease—such as females younger than 60 years [70]. In cases of suspected CTD-ILDs, further investigation should be carried out based on the clinical suspicion of a rheumatologist, which may involve additional tests such as biochemical screening (including autoantibodies), capillaroscopy, or ultrasound. Another scenario involves ILD patients with indeterminate HRCT UIP patterns, which are frequently observed during CTD. In such cases, before performing additional tests such as capillaroscopy or ultrasound, a screening test including ANA, RF, ACPA, and creatine phosphokinase (CPK) dosage should be performed [71].

Another diagnostic challenge arises with IPAF, which is a relatively newly characterized nosological entity. IPAF refers to an ILD where clinical or serological abnormalities typical of CTD are present but insufficient to satisfy the classification criteria of a defined autoimmune disease. Without a rheumatologic evaluation, there is a risk of misclassifying these patients [72]. The importance of an MDA is also expressed by the same definition of IPAF. The diagnosis of IPAF is based on radiological and/or histological identification of ILD in the absence of precise etiology, having excluded other CTDs, and presenting at least two of three clinical, serological, or morphological domains [73]. This recent proposal of classification criteria for IPAF represents an effort to underline the need to involve a rheumatologist in the MDT, especially in helping to diagnose lung involvement associated with a clear underlying CTD or with autoimmune features [13,74].

## 9. MDT Composition and Timing

Despite the importance of MDD, there are no indications about the optimal composition of the expert team or the timing of these meetings. Typically, the MDT for ILDs includes a pulmonologist, a radiologist, and a pathologist, all of whom are experts in ILDs. Depending on the cases, other physicians like specialists in rheumatology, thoracic surgery, lung transplantation, and occupational medicine can be involved. The level of expertise among participants determines the frequency of these meetings [46]. The current gold standard for ILD diagnosis is a dynamic integrated process that requires direct interaction between clinicians, radiologists, and pathologists when an SLB is available [11].

A multidisciplinary approach to palliative care, involving ILD experts, a palliative respiratory care expert, a nurse, a respiratory therapist, a physiotherapist, and a dietitian is effective compared with the standard approach, which typically involves ILD experts and a nurse only. This comprehensive approach results in improved efficacy in the management of patients, in terms of a reduced number of emergency visits and hospital admissions [59].

Another substantial difference in clinical practice, compared with the ERS/ATS guidelines [16], is the role of the rheumatologist during ILD evaluation. According to these guidelines, the presence of a rheumatologist in MDD meetings is not mandatory for every newly identified ILD. Instead, their involvement is restricted to cases with positive autoantibodies tests, suspected CTD clinical manifestations, or in cases with unusual features of IPF, such as female sex, age younger than 60 years, and absence of tobacco habit. The engagement of a rheumatologist expert among MDD participants could be useful to identify systemic clinical manifestations, particularly in patients with clinical and histopathological features inconsistent with IPF. Even though one in five ILD cases can be related to CTD, a systematic evaluation [75] of diagnostic ILD practices conducted in 2019 showed that only 37.1% of MDT cases include a rheumatologist, indicating a significant impact on the final diagnosis. Despite current IPF guidelines, the involvement of a rheumatologist in the diagnostic process is entrusted by the experience of the individual pulmonologist. A study by De Lorenzis et al. [76] emphasized the importance of a collegial discussion of challenging cases of CTD-ILDs. Such discussions when including a rheumatologist could add some important details overlooked by the evaluation of the pulmonologist alone. This multidisciplinary approach can increase confidence in the diagnosis and enhance the therapeutic management of the patient.

Chartran’s retrospective study [77] highlighted the role of the rheumatologist in the MDT for ILD patient evaluation. The study found that the initial IPF diagnosis was often changed after MDD, particularly after the identification of specific autoimmune antibodies. The authors underlined the need to extend the screening of autoimmune profiles because about one-third of the patients were ANA-negative and not all cases presented systemic manifestations. Another retrospective observational study [60] involving 50 patients showed that MDD led to a final diagnosis of CTD-ILD in 25 patients, 7 of whom were initially identified as IPF with completely different prognostic and therapeutic implications. In a prospective study [61] of 60 patients, where the MDT also included a rheumatologist, 21.9% of IPF cases and 28.5% of HP cases showed their diagnosis modified, favoring CTD disease and avoiding invasive procedures such as bronchoscopies and lung biopsy. Despite the evidence in the literature, there is a lack of clear guidance on the timing of rheumatologist involvement during multidisciplinary meetings, in particular during ILD-CTD or IPAF evaluation.

Although MDD represents the gold standard in ILD diagnosis, it is not always practicable in a clinical routine because of a lack of skilled experts or difficulties in organizing meetings due to the geographical distance between the participants or time-related limitations. Fujisawa et al. [78] emphasized the role of validated digital platforms, which allow easier access to various data via the Web. Their study involving 465 ILD patients showed that MDD could reformulate the initial diagnosis in 49% of cases. Table 2 provides an overview of the progression and development of the MDD concept over the years according to different guidelines.

## 10. Conclusions

According to the common symptoms and frequent overlapping of radiological and/or histological data, the importance of multidisciplinary management of ILDs has gradually become more evident over the years. Based on the integration of clinical, radiological, histopathological, and often serological data, an MDA improves the accuracy and reliability of ILD diagnosis, reducing the need for invasive procedures such as SLB or TBLC. In challenging cases that require histologic data, such as when faced with the presence of an indeterminate or inconsistent HRCT UIP pattern, the MDT plays a crucial role in selecting the most suitable technique (SLB or TBLC) with the best diagnostic yield and minimal invasiveness, thereby reducing the risk of side effects. After the evaluation of integrated clinical and radiological elements, TBLC is suggested as the first option in central lesions with compromised lung functions only in centers with high expertise, while SLB remains the gold-standard technique for peripherical lesions and preserved lung function. Further studies are required to evaluate the diagnostic yield and the risk profile associated with awake/non-intubated VATS biopsy.

The MDD of ILDs should not be considered a static process, but a working diagnosis, based on the progressive integration of clinical, radiological, and pathological data available, allowing for the initiation of a specific therapy even with a diagnosis of probability. The periodic evaluation of the disease progression and the response to therapy increase the accuracy of diagnosis, confirming or modifying the therapy and ensuring optimal patient care and outcomes.

Since its institution for IPF diagnosis due to its high level of diagnostic confidence and inter-observer agreement, MDD has progressively expanded to other ILDs, replacing histopathology as the diagnostic gold standard. In recent years, the literature has emphasized the importance of an MDA in the diagnosis of ILDs other than IPF, such as HP or CTD-ILDs, defining the composition of the MDT. Notably, the absence of evidence-based diagnostic guidelines in non-IPF ILDs contributes to a certain level of disagreement in MDD. In this context, more studies are required to confirm the role of MDD in the diagnostic and therapeutic management of ILDs.

## Figures and Tables

**Table 1 diagnostics-13-02437-t001:** The impact of MDD on changing the final diagnosis.

Study	Change in Diagnosis after MDD
Aziz et al. [43]	51.0%
Ageely et al. [47]	37.0%
Kondoh et al. [15]	50.0%
Jo et al. [54]	53.0%
De Sadeleer et al. [46]	41.9%
Nakamura et al. [57]	45.5%
Thomeer et al. [42]	12.8%
Han et al. [53]	10.7%
Castelino et al. [60]	28.0%
Levi et al. [61]	50.4%

MDD: Multidisciplinary discussion.

**Table 2 diagnostics-13-02437-t002:** Evolution of the MDD process according to guidelines.

Document	Year	ILD Type	Changes in the Diagnostic Process
ATS/ERS Multidisciplinary consensus classification of the IIPs [5]	2001	IIP	MDD: for IPF replacement of histology via the integration of clinical and radiological typical data MDT: pulmonologists, radiologists, and pathologists Aim: diagnosis as a dynamic process
An official ATS/ERS statement: update of the international multidisciplinary classification of the IIPs [13]	2013	IIP	MDD: in all ILDs, in particular in differencing NSIP from HP, in coexisting patterns, in rare histologic patterns, and unclassifiable IIP Aim: when to perform a biopsy; disease behavior
An official ERS/ATS research statement: IPAF [11]	2015	IPAF	MDD: integration of clinical, serological, and morphological domains MDT: clinic, radiologist, pathologist, and rheumatologist
Diagnostic criteria for IPF [14]	2018	IPF	MDD: in case of atypical characteristics or features related to non-IPF etiology MDD: clinician, radiologist, and pathologist; rheumatologist can be helpful MDT timing and place: weekly/monthly; direct or telemedicine Aim: diagnosis, management, and revaluation over time
Diagnosis of HP in Adults: An Official ATS/JRS/ALAT Clinical Practice Guideline [63]	2020	HP	MDD: in patients with atypical exposure history, radiological features, or cellularity on BAL; histopathological sampling should be made after MDD in low/moderate confidence diagnosis MDT: clinician, radiologist, and pathologist Aim: diagnosis and re-evaluation based on additional clinic/pathologic data
Diagnosis and Evaluation of HP: CHEST Guideline and Expert Panel Report [64]	2021	HP	MDD: BAL is not mandatory in the diagnosis, but should be performed after MDD in patients with low/moderate test probability of HP; lung samples should be made after MDD in atypical cases Aim: confident diagnosis avoiding invasive procedures
Diagnosis and management of CTD-ILDs in Australia and New Zealand: A position statement from the Thoracic Society of Australia and New Zealand [68]	2021	CTD-ILD	MDD: face-to-face discussion between experts integrating clinical, radiological, histopathological, and serological data MDT: respiratory physicians, radiologists, pathologists, immunologists, and rheumatologists Aim: confidence diagnosis avoiding invasive procedures
IPF (an Update) and Progressive Pulmonary Fibrosis in Adults: An Official ATS/ERS/JRS/ALAT Clinical Practice Guideline [16]	2022	IPF and PPF	MDD: IPF diagnosis for UIP or possible UIP HRCT pattern in the appropriate clinical setting without confirmation via lung biopsy Aim: diagnosis with less invasive procedures and working diagnosis

MDD: multidisciplinary diagnosis, ILD: interstitial lung disease, ATS: American Thoracic Society, ERS: European Respiratory Society, IIP: idiopathic interstitial pneumonia, IPF: idiopathic pulmonary fibrosis, MDT: multidisciplinary team, NSIP: non-specific interstitial pneumonia, HP: hypersensitive pneumonia, IPAF: idiopathic pneumonia with autoimmune features, JRS: Japanese Respiratory Society, ALAT: Latin American Thoracic Society, BAL: bronchoalveolar lavage, CTD: connective tissue disease, UIP: usual interstitial pneumonia, HRCT: high-resolution chest tomography.

## Data Availability

No new data were created or analyzed in this study. Data sharing is not applicable to this article.

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
