# Peer review of "The Evolving Concept of the Multidisciplinary Approach in the Diagnosis and Management of Interstitial Lung Diseases"

_diagnostics, 2023, doi:10.3390/diagnostics13142437_

Round 1
Reviewer 1 Report
In their review regarding the value of the multidisciplinary team (MDT) in ILDs, the authors do not manage to condensate and present to readers the current scientific evidence concerning MDT. Indeed, the importance of MDT in the prompt diagnosis and prognosis of IPF and ILDs in general is well defined and recognised globally. The manuscript is not well organised not permiting to readers to have safe conclusions about the most recent scientific data regarding its role. Therefore, i am concerned. So, please reorganise the text. Also, use more Tables and figure to shape the current evidence.
Also, there is no mention in the latest aspects including genomic classifier or cryobiopsy where MDT may have an important role.
Author Response
The latest version of the review included new topics such as the role of MDD in choosing cryo-biopsy or surgical biopsy, the effect of genomic classifier testing on the MDD of ILDs, and the diagnosis of HP and CTD-ILDs based on the new guidelines. Substantial changes have been made to the English used in this review and in the global organization. Also, a new table regarding the change in the concept of MMD was added to the text.
Reviewer 2 Report
The manuscript tries to highlight the added value of the MDD/MDT in ILD management. Despite the few data available in the literature, which are partially discussed in the text, there is a big gap of knowledge.
Given that Diagnostics is not a Pulmonology journal, I think the manuscript would benefit from radiology, rheumatology and pathology inputs, to make it more appealing to a broader spectrum of readers. The presentation of some difficult cases, "resolved" through a MDD could further improve the paper.
In addition, the authors could provide proposals for improving the areas where there is lack of knowledge, such as check-lists for referral, proposals of shared decision making process, how an MDD could be structured. This would be of added value in this manuscript which, otherwise, has little sound.
Style and English require substantial revision.
In addition, abbreviations should be clarified when they appear in the text, even if already presented in the abstract.
The manuscript looks written in Italian and then translated. Certain words which make sense in Italian have a complete different meaning in English. Articles are very frequently missing or used wrongly, many spelling errors. The construction of the sentence is very "latin".
It requires grammar check and a thorough improvement in style.
Author Response
In the latest version of the review were added more topics concerning diagnostic aspects such as the role of MDD in choosing cryo-biopsy or surgical biopsy, the effect of genomic classifier testing on the MDD of ILDs, and the diagnosis of HP and CTD-ILDs based on the new guidelines. The title of the paper has been changed without transforming the meaning. Substantial changes have been made to the overall organization of the text and to the English used in this review. Also, a new table was added to the text.

Reviewer 3 Report
Although the paper addresses the topic of MDD approach , it does not bring anything new for the reader familiar with interstitial pathology.
For young specialists, without expertise in the ILD field, reading the article could be useful. Perhaps publication in a less demanding journal would be more appropriate.
Author Response

(The authors gave the same response as above.)

Round 2
Reviewer 1 Report
The article does not achieve to present recent advances in the field. There are also issues in the organisation and presentation of the text. Authors reply is not sufficient to cover the proposed modifications.
Author Response
Thank you for your comments and suggestions. According to the previous report, I improved the text with better organization including a new table (lines 483-488) and mentioning required aspects such as the role of MDD in choosing the biopsy sampling technique (lines 107-201) and genomic classifier (lines 203-236).

Reviewer 2 Report
The flow of the text has well improved.
some typos:
line 111, liked--> is it "linked"?
line 334: LID stands for ILD?
line 584: AST --> ATS
line 618: remove "cutaneous"
line 646: echography --> ultrasound
Moreover, the conclusions should be shortened, as they look more like a summary of the whole manuscript.
improved
Author Response
Thank you for your comments and suggestions. As asked I shortened the conclusion (lines 489-513) and corrected the typos reported such as:
linked instead of linked (line 68)
ILD instead of LID (line 288)
ATS instead of AST (line 374)
I removed the word cutaneous (line 396)
ultrasound instead of echography (lines 424 and 426)

Reviewer 3 Report
In the last review paper, the authors added more references and structured better the literature. the structure of the article is more clear, and with important changes. Maybe some comments about the role of surgery /cryo-biopsy in the context of antifibrotic for progressive phenotype could help better the young practician, especially where the MDD is not available.
With minor revisions, the article could be published.
Author Response
Thank you for your comments and suggestions. As requested, I shortly commented on the role of surgery and cryo-biopsy in the context of antifibrotic therapy for PF-ILDs (lines 182-201).

Round 3
Reviewer 1 Report
Unfortunately, i would suggest the paper be rejected. As i had stated in my first review, the authors do not manage to present novel insights in the field.